# Designing the Location–Routing Problem for a Cold Supply Chain Considering the COVID-19 Disaster

**Sina Abbasi [1], Maryam Moosivand [2], Ilias Vlachos [3],* and Mohammad Talooni [4]**

[1] Department of Industrial Engineering, Lahijan Branch, Islamic Azad University, Lahijan 1616, Iran; abbasisina170@gmail.com or abbasisina@liau.ac.ir

[2] Institute for Management and Planning Studies, Tehran 1411713135, Iran; maryam_musivand69@yahoo.com

[3] Excelia Business School, Excelia Group, 17000 La Rochelle, France

[4] School of Industrial Engineering, College of Engineering, University of Tehran, Tehran 141556311, Iran; mohamad.talooni@ut.ac.ir or mohamad.talooni@gmail.com

* Correspondence: ivlachos@gmail.com or vlachosil@excelia-group.com; Tel.: +335-4651-7700

**Abstract:** In this study, a location routing problem (LRP) model was considered for the distribution network of multiple perishable food items in a cold supply chain (CSC) where vehicles can refuel at gas stations during light of the COVID-19 disaster. Fuel consumption is assumed to vary depending on the cargo transported between nodes when using a non-standard fuel fleet. The problem was formulated as a mixed-integer linear programming (MILP) model to reduce the production of carbon dioxide ($CO_2$). The model was validated using several numerical examples which were solved using the software, LINGO 17.0. The results show that fuel consumption could be reduced in this case. Due to the complexity of the problem, genetically simulated annealing algorithms were developed to solve the actual size problems, and their performance was also evaluated.

**Keywords:** sustainable supply chain; COVID-19 disaster; cold logistics; $CO_2$ emissions; location routing problem; genetic algorithm

## 1. Introduction

The spread of the new coronavirus disease (COVID-19) threatens the health of the world's population, and it is the latest epidemic that has led to a pandemic and a public health emergency [1]. By the last day of March 2020, COVID-19 had spread rapidly from Wuhan to other areas, and it had affected more than 200 countries worldwide [2]. Due to the sharp increase in the number of people affected, and the initial lack of attention paid to the COVID-19 pandemic by world leaders, it became a global epidemic [3].

The negative impact of automobile traffic on the environment is undeniable due to its impact on the earth and the consumption of resources. The emission of greenhouse gases has led to an increase in average global temperatures, which is a well-known phenomenon called global warming [4]. Moreover, the efficient management and optimization of logistical activities has led to offering customers different and faster services, reduced waiting times, reduced damages, and improved customer care, taking the critical role of transport in generating and releasing biological pollutants in the environment into consideration. These pollutants have destructive and irreparable effects on the environment, the health of living beings, and the ozone layer. Considering sustainability, environmental constraints, and economic imperatives when modeling and optimizing transportation and distribution networks will significantly help preserve the environment and reduce associated risks [5]. Vlachos and Malindretos (2023) [6] redesigned the SC for aquaculture with a real case study.

Perishable food LRP designs optimize location, inventory, and routing decisions in a supply chain system (SC) [7]. The goal is to minimize costs, reduce food waste, and improve the efficiency and sustainability of the SC [8]. Research papers and articles provide

insights into approaches and models for solving the LRP for perishable products. Here are some key points from the search results.

The LRP for perishable products involves the integration of location, inventory, and routing decisions in a SC system [7,9]. This integration is critical for optimizing the overall performance of the SC [10].

When designing the location routing problem (LRP), minimizing the total cost, including transportation, inventory, and food waste costs, is a key aim [9,11]. Optimizing the total cost formula helps to achieve cost-effective operations. Designing a sustainable supply chain network (SSCN) for perishable foods is important [12]. This requires optimizing the economic and environmental aspects of the SC [13]. Strategies such as dynamic pricing and minimizing $CO_2$ emissions can be used to improve resilience and sustainability [11,14]

Vlachos and Polichronidou [15] (2023) suggested a model for multi-demand, with regard to SC issues. Shafaghizadeh et al. (2021) [16] analyzed the role of wholesale markets in food SCs by considering their resilience. Meta-heuristic algorithms, such as genetic algorithms, can be used to solve the LRP for perishable products. These algorithms provide efficient solutions within an acceptable time frame [17]. The LRP for perishables is a special case concerning the vehicle routing problem (VRP) [18]. The VRP involves optimizing routes and schedules for vehicles delivering perishable products, considering time constraints and quality degradation [19].

These insights from the search results provide a starting point for designing the LRP for perishable foods. Further research and analysis can help develop specific models and algorithms tailored to the unique requirements of the perishable food industry [20]. Li et al. (2018) [21] suggested a multi-agent for the SC framework in light of uncertain demands. Li et al. (2023) [22] proposed a SC for the partial durable good. [23] Xiao et al. (2021) reviewed a comprehensive model with nonlinear energy recharging and consumption. Huang et al. (2023) [24] improved blueberry freshness predictions using machine learning for cold chain logistics.

Figure 1 shows the transportation process for refrigerated logistics. The different sections of the paper are organized as follows: Section 2 reviews previous studies, Section 3 presents the problem, and Section 3.2 illustrates the linearization of the mathematical model of the problem. Section 4 presents the solution, Section 5 presents the validation of the mathematical model and its solution using genetic and simulated annealing algorithms, and Section 6 presents the conclusions and suggestions for future research.

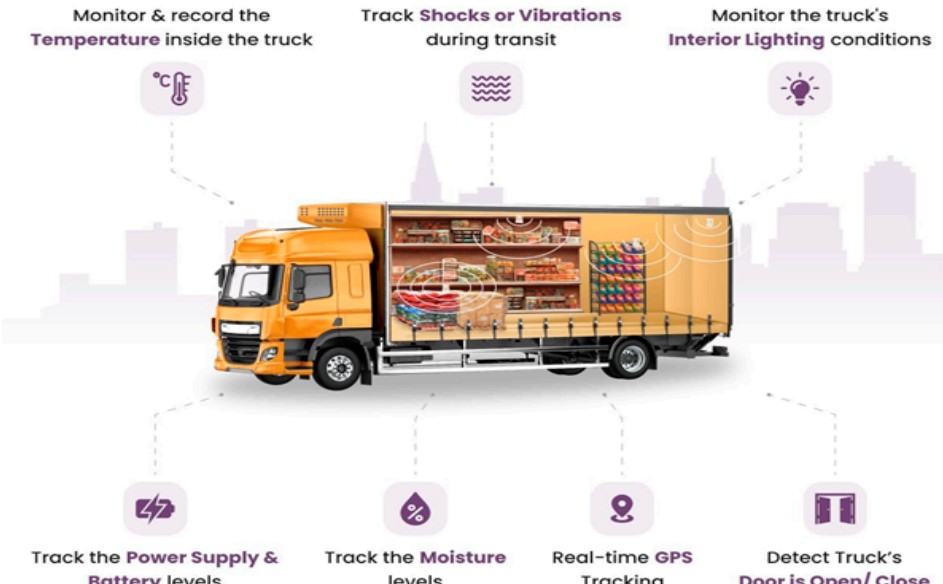

**Figure 1.** With the ability to monitor and manage the transportation process in real-time, IoT technology can solve problems, especially those concerning perishable goods [25].

## 2. Literature Review

### 2.1. Related Studies

The impact of the COVID-19 pandemic on LRP is currently the subject of extensive research, although not at the level of deployment. In addition to providing medical items, another important issue concerns managing infectious medical waste associated with COVID-19 after diagnosing and treating patients in health centers, including hospitals and dispensaries. As the number of confirmed cases has increased, the amount of medical waste associated with COVID-19 has increased significantly, and it is now considered a critical hazardous material. In other words, medical waste disposal is considered an important measure to control the source of infection, and it is necessary to strictly define and standardize the waste disposal of COVID-19 [26].

However, since most of this waste is made of plastic, it can be hazardous to the environment if not processed properly and promptly. Therefore, to the greatest extent possible, careful attention should be paid to reducing the risk of epidemics in hospitals and hospital wards. In this study, a waste management system is designed to efficiently handle collection (from hospitals and hospital wards), transportation (via the road network), and the disposal of COVID-19 waste at pre-determined disposal sites [27]. This section briefly explains the routing and location problem, and then, the main studies related to this research are briefly reviewed. Ahmed and Yousefi Khoshbakht (2023) [28] propose that VRP refers to problems where a fleet of multiple vehicles, from one warehouse, provides customer services at different locations.

Due to the limited loading capacity of vehicles, the VRP is sometimes referred to as the capacitated VRP. The VRP is divided into several types based on its limitations and special conditions. The multi-depot vehicle routing problem (MDVRP) is one of these types. The MDVRP is an evolved version of the classical VRP, where the number of warehouses is more than one. In this problem, customers are first assigned to the existing warehouses using techniques such as clustering. After assigning the customers to the warehouses, the transportation routes between each warehouse and the customers are determined under the constraints of the classical VRP. Moreover, the problem of facility location is also one of the areas considered in operations research. In a general location model, facilities are scattered in a geographical area to satisfy customer demand [29].

Location models have been extended to answer the abovementioned questions with different objectives and assumptions, which has led to the emergence of different types of location models with extensive scopes. However, a question arises concerning the relationship between location models and routing problems. In response, the link between location and routing problems is explained via the design of distribution and logistics networks [30].

The issues at the strategic level of logistics engineering and supply chain management (SCM) relate to the number of distribution warehouses and their locations, the customers assigned to each distribution center, the number of vehicles that deliver customer orders and their routes, etc. If location and route planning are not considered simultaneously, supporting SC costs will increase [31–36]. Bektas and Laporte (2011) [37] considered pollution in the vehicle routing problem and presented their proposed model. Liu and Yu (2012) [38] presented the multi-depot vehicle routing problem based on an ant colony with a genetic algorithm. Erdogan and Hooks (2012) [39] presented a green vehicle routing problem which they solved using Clark and Wright's density-based clustering algorithm and an improved heuristic method. Xiao et al. (2012) [40] presented a model for capacitated LRP in which the cost of fuel consumption was considered, in addition to other costs, to optimize the total cost and reduce environmental pollutants. The model used two numerical examples available in the reviewed literature, and the responses showed very different energy consumption levels, with regard to the classical vehicle routing problem, compared with the energy consumption of the load-dependent condition.

Zheng and Chen (2014) [35] presented an optimization model for a multi-product frozen food vehicle routing problem, taking delivery time into consideration, and they

solved the problem using the genetic meta-heuristic method. Goeke and Schneider (2015) [41] modeled the mixed fleet vehicle routing problem with two types of vehicles, and they considered the amount of fuel and energy consumed in conjunction with speed, road slope, and weight of cargo. Moreover, they noted the refueling capabilities of electric vehicles, and then, they developed an innovative local search solution method. Koc et al. (2016) [42] proposed to model the LRP with a heterogeneous transportation fleet and a time window, first as a mixed integer programming problem, and then as a family of related constraints. They attempted to solve this problem by developing a powerful algorithm, a hybrid evolutionary algorithm.

Bae and Moon (2016) [43] introduced the multi-warehouse vehicle routing problem with a time window, and they described delivery and setup times for heterogeneous vehicles to minimize the total cost. Song and Ko (2016) [44] presented the vehicle routing problem, with two types of normal and glowing vehicles for perishable food, to maximize customer satisfaction in terms of maintaining freshness. Xiangguo and Manying (2015) [45] developed a route to solve a cold chain logistical problem, which they combined with a time window to satisfy multiple customers. They achieved this by assuming the probability of each customer's demand, and then they solved the problem using a genetic algorithm. Keskin and Catay (2016) [46] introduced a routing problem for electric vehicles, combined with a time window; this problem is an extension of the well-known classical time window problem for vehicle routing, and they adapted an adaptive algorithm for searching in large neighborhoods to solve this problem efficiently.

Wang et al. (2016) [47] presented a Multi-Objective Vehicle Routing Problem (MOVRP), with a time window for perishable food distribution, and they aimed to minimize the total cost and maximize the freshness of the products delivered to customers. They solved this problem with a two-phase heuristic algorithm, based on the Pareto variable neighborhood search and Genetic Algorithm, by describing the space–time interval. Montoya et al. (2017) [48] formulated a routing problem for electric vehicles with a nonlinear charging function. Schiffer and Walther (2017) [49] presented the electric vehicle routing problem with time windows and partial charging. Hosoda and Irohara (2022) [50] proposed an LRP approach to describe electric vehicle routing and charging station decisions in order to support the strategic decisions of the operational logistics fleet simultaneously. Wu et al. (2017) [51] attempted to design an integrated distribution system for food services on high-speed trains based on the three principles of location, routing with hard time windows, and deadlines. They designed this system in order to deliver high-quality perishable food on trains, taking demands that are influenced by various aspects of railroad planning into consideration. They solved this problem using a hybrid cross-entropy algorithm. Hsiao et al. (2017) [52] modeled a distribution scheduling problem for a cold food chain to generate a distribution plan that satisfied customer needs for a variety of foods with a pre-determined quality level at the lowest distribution cost. To solve the problem, a biography-based compatibility optimization algorithm with genetic algorithm modeling was used. An article by Wang et al. (2018) [53], titled Optimization of the Location Routing Problem (LRP) for the Cold Chain Logistics (CCL), concerns the carbon phenomenon, and it refers to the category of environmental protection that examines how carbon minimization can minimize the total cost, which includes the carbon emission cost. They also developed a hybrid genetic algorithm with heuristic rules to solve the model. Simulation results of a practical numerical example showed the model's applicability to provide distribution plans and green and nature-friendly sites for CCL companies. Table 1 shows the main features of the research papers discussed in this section.

**Table 1.** The perspective of the current study with regard to previous studies.

| Researchers' Names | Warehouse Number | | Type of Fleet | | Decay | | Time Window | Length of Tours | | Product Variety | Type of Fuel Used | | Type of Fuel Consumption | | Tank/ Battery Capacity | Refueling Capability | Disaster |
|---|---|---|---|---|---|---|---|---|---|---|---|---|---|---|---|---|---|
| | One | Multiple | Heterogeneous | Homogenous | Refrigeration | Deterioration | | Time | Distance | | Electric | Combustion | Constant | Variable | | | |
| Bektas and Laporte (2011) [37] | ✔ | | | ✔ | | | ✔ | | | | | | ✔ | | ✔ | | | |
| Liu and Yu (2012) [38] | | ✔ | | ✔ | | | ✔ | | ✔ | | ✔ | | | | | | | |
| Erdogan and Hooks (2012) [39] | ✔ | | | ✔ | | | ✔ | ✔ | | | | | ✔ | ✔ | | ✔ | ✔ | |
| Xiao et al. (2012) [40] | ✔ | | | ✔ | | | | | | | | | ✔ | ✔ | | | | |
| Zheng and Chen (2014) [35] | ✔ | | | ✔ | ✔ | ✔ | ✔ | | | ✔ | | | | | | | | |
| Goeke and Schneider (2015) [41] | ✔ | | ✔ | | | | ✔ | | | | | ✔ | ✔ | ✔ | ✔ | ✔ | | |
| Koc et al. (2016) [42] | | ✔ | ✔ | | | | ✔ | | | | | | | | | | | |
| Bae and Moon (2016) [43] | | ✔ | ✔ | | | | ✔ | ✔ | | | | | | | | | | |
| Song and Ko (2016) [44] | ✔ | | ✔ | | ✔ | | | ✔ | | ✔ | | | | | | | | |
| Xiangguo and Manying (2015) [45] | ✔ | | | ✔ | ✔ | ✔ | ✔ | | | | | | | | | | | |
| Keskin and Catay (2016) [46] | ✔ | | | ✔ | | | ✔ | | | | | ✔ | | ✔ | | ✔ | ✔ | |
| Wang et al. (2016) [47] | ✔ | | | ✔ | ✔ | | ✔ | | | | | | | | | | | |
| Montoya et al. (2017) [48] | ✔ | | | ✔ | | | | ✔ | | | | ✔ | | ✔ | | ✔ | ✔ | |
| Schiffer and Walther (2017) [49] | ✔ | | | ✔ | | | ✔ | | | | | ✔ | | ✔ | | ✔ | ✔ | |
| Wu et al. (2017) [51] | | ✔ | | ✔ | | | ✔ | ✔ | | ✔ | | | | | | | | |
| Hsiao et al. (2017) [52] | ✔ | | | ✔ | | | ✔ | | | ✔ | | | | | | | | |
| Wang et al. (2018) [53] | | ✔ | ✔ | | ✔ | ✔ | ✔ | | | | | ✔ | ✔ | | ✔ | ✔ | ✔ | |
| Current Research | | ✔ | ✔ | | ✔ | ✔ | ✔ | | | | | ✔ | ✔ | | ✔ | ✔ | ✔ | ✔ |

### 2.2. Research Gap Analysis and Contributions

A review of the literature shows that, so far, extensive research has been conducted at a series of locations, and on numerous routing problems. However, simultaneous investigations into multi-storage modes, the use of heterogeneous vehicles with non-conventional fuels, variable fuel consumption rates (which vary in accordance with the size of the load), the refueling ability of different types of vehicles, multi-product modes, and perishability for the CSC have not been conducted with the COVID-19 disaster in mind.

### 3. Problem Presentation

In this study, the LRP for CSC is considered. The critical points are fuel consumption, the size of the load, and the COVID-19 catastrophe. The graph $G = (V', A)$ contains the set $V'$ of all nodes, including the fixed locations of customers, the potential locations of warehouses, and the potential locations of fueling stations. A set of edges is defined below, including parameters and decision variables. This mathematical model minimizes harmful environmental impacts by considering the fuel consumption between nodes during the COVID-19 disaster.

### 3.1. Mathematical Modeling

In this section, first, the assumptions of the proposed mathematical model are explained, then, the sets, parameters, and decision variables are presented (Tables 2–4). Finally, the mathematical model is presented.

**Table 2.** The sets of models.

| Symbol | Description |
|---|---|
| C = {1, 2, ..., c} | The set of customers |
| G = {1, 2, ..., g} | The set of warehouses |
| P = {1, 2, ..., p} | The set of cold food products |
| T = {1, 2, ..., t} | The set of vehicles |
| F = {1, 2, ..., f} | The set of refueling places |
| F' = {1, 2, ..., F + 1, F + 2, ..., F + F'} | The set of virtual charging and refueling places |
| V = C∪F' | The set of customers and virtual charging and refueling places |
| V'' = F'∪G | The set of warehouses and virtual charging/refueling places |
| V' = V∪G = G∪F'∪C | The set of customers, virtual refueling places, and warehouses |

**Table 3.** The parameters of the model.

| Symbol | Description | Symbol | Description |
|---|---|---|---|
| $Pr_p$ | The price of a unit of a cold production type $p$ | $EST_c$ | The earliest service start time at the $c$ node |
| $\mu_p$ | COVID-19 damage rate of cold production type $p$ during customer service time | $LST_c$ | The latest service start time at the $c$ node |
| $\theta_p$ | COVID-19 damage rate of cold production type $p$ during the transportation process | $Q_t$ | The capacity of vehicle $t$ |
| $U_{pc}$ | The quantities of cold production type $p$ by the customer $c$, Zero if c $\notin$ C | $d_{cg}$ | Distance between nodes $g$ and $c$ |

**Table 3.** *Cont.*

| Symbol | Description | Symbol | Description |
|---|---|---|---|
| $\beta$ | The frequency of opening and closing the vehicle's refrigerator door to serve the customer | $r_t$ | The refueling rate of vehicle $t$ |
| $B_t$ | The capacity of the fuel of vehicle $t$ in terms of the fuel unit | $s_c$ | Service time at the customer $c$ |
| $\alpha_0^t$ | The fuel consumption rate of the vehicle $t$ in its unloaded state in units of fuel per kilometer | $TR_{cg}$ | Travel time between nodes $g$ and $c$ |
| $\alpha_t^*$ | The fuel consumption rate of the vehicle $t$ in its maximum load mode in fuel per kilometer | $CF_t$ | The fixed cost of using the vehicle $t$ |
| $\rho_{cg}^t$ | The fuel consumption rate of vehicle $t$ between nodes $g$ and $c$ | $F_g$ | The fixed cost of building the warehouse $g$ |
| $fp_{cg}^t$ | The amount of fuel consumption of vehicle $t$ between nodes $g$ and $c$ | $Q_g$ | The maximum capacity of warehouse $g$ |
| $C^t$ | The cost of a unit of fuel for a vehicle $t$ | $O_p$ | The weight of a unit of cold food type $p$ |
| $\omega_t$ | The depreciation coefficient of the refrigerators of vehicle $t$ | $Q_t$ | The capacity of vehicle $t$ |
| $\delta$ | The coefficient of the temperature conductivity of the refrigerator's body | $\propto$ | A very large positive number |
| $\Delta T$ | The temperature difference between the outside and inside of the vehicle's refrigerator | $\sum S_n$ | The total external surfaces of refrigerators in vehicle |
| $\sum S_w$ | The total internal surfaces of the refrigerators in the vehicle | $\Gamma_\tau$ | The thermal load related to the temperature difference between the outside and inside of the refrigerator |
| $\Gamma_\sigma$ | The thermal load during loading and unloading time | | |

**Table 4.** The decision variables of the model.

| Symbol | Description |
|---|---|
| $\tau_g^t$ | The arrival time of vehicle $t$ at node $g$ |
| $F_{cg}$ | The number of transported goods between nodes $c$ and $g$ |
| $y'_{gh}$ | The remaining capacity for fuel in vehicle $t$ when entering node $g$ |
| $y''_{gt}$ | The remaining capacity for fuel in vehicle $t$ when entering node $g$ |
| $X_{cgt}$ | If the movement from node $c$ to node $g$ by vehicle $t$ is complete, it equals 1; otherwise, it equals zero |
| $W_{cg}$ | If customer $c$ is assigned to warehouse $g$, it equals 1, otherwise, it equals zero |
| $M_g$ | If the warehouse $g$ is opened, it equals 1; otherwise, it equals zero |
| $X'_{cgt}$ | The auxiliary binary decision variable for the constraint linearization of constraints |

### 3.1.1. Model Assumptions

- There are different types of vehicles with internal combustion engines.

- The customers' locations, service stations, and distribution centers (warehouses) are specific.
- The demand of each customer is obvious. Each customer has a specific, pre-determined delivery time for his or her order.
- The fuel consumption of the vehicles is variable and depends on the amount of goods transported between the two nodes.
- The travel time between node $j$ to node $i$ is calculated using cars. The time to serve the customers and refuel at the stations is specified for each type of car.
- Each customer's order is delivered by only one car, but it can serve different customers.
- Each customer's demand does not exceed the capacity of each group of vehicles. The vehicles have a fixed speed. Each vehicle is only allowed to refuel once at a service station.
- Each group of vehicles has a specific and fixed refueling capacity.
- At the fueling stations, the waiting time in the queue will be zero.
- Each tour starts at one of the reopened warehouses and it ends at the same warehouse.

The model's sets, parameters, and decision variables are expressed above.

Fuel Consumption Estimation Models

Fuel consumption costs have always accounted for a large part proportion of transportation costs, and fuel consumption causes an escalation in terms of the production of greenhouse gases, the release of environmental pollutants, and an adverse effect on the ecosystem. Therefore, paying attention to planning, and optimizing logistical activities using mathematical models, has become necessary. This section will explain how to calculate and determine the rate and amount of fuel used for car engines when moving between two nodes [54].

A Model for Calculating the Fuel Consumption of Cars When Moving

The model in Figure 1 was used to calculate the required fuel consumption of a car engine that advances and moves between vertices.

Formulation of the Car Engine Fuel Consumption Rate

Although the amount of fuel consumption is determined based on traveled distance, to a large extent, other parameters, such as the amount of transported cargo, are also effective in terms of reducing the cost of fuel consumption. According to the report published by the Ministry of Road Organization concerning the transport and tourism infrastructure of Japan, travel distance and units of fuel used to have a strong relationship and correlation with increases in vehicle weight (see Figure 2). In Figure 2, the red line (with a square marker) represents the actual fuel consumption rate, which is shown to be proportional to an increase in car weight, resulting from the statistical data. The blue line (with a circle marker) shows the linear regression from this data, which is formulated as $Y = 0.08X - 0.03$, and since the indicator for matching the prediction with real data values is equal to 90.2%, the blue line regression function can confirm the general relationship between the fuel consumption rate and an increase in vehicle weight.

Figure 2 presents the fuel consumption rate in accordance with the gross vehicle weight. By generalizing this topic and dividing the gross weight of the vehicle into two parts, including unlade weight ($\theta_c$) and loaded weight ($\theta_u$), the fuel consumption rate formula can be approximately presented in fuel units per kilometer as a linear function, depending on the weight of the loaded cargo $\alpha(\theta_u)$; where $\omega$ is the angular coefficient and b is the intercept of the regression function related to determining the car's fuel consumption rate.

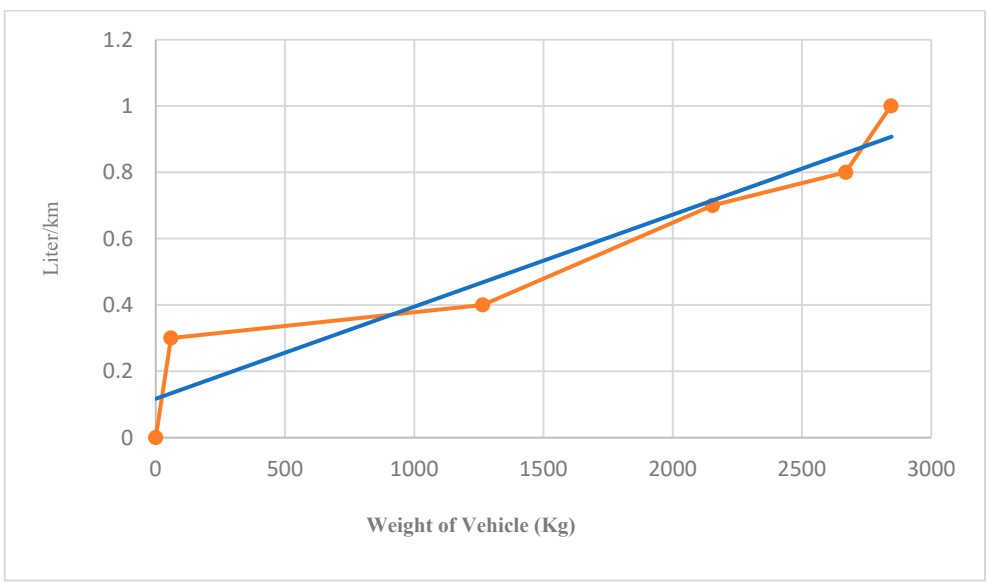

**Figure 2.** Fuel consumption rate based on gross vehicle weight.

$$\alpha(\theta_u) = \omega(\theta_c - \theta_u) + b \tag{1}$$

If the maximum capacity of vehicle $t$ (maximum loadable weight) is $Q_t$, the fuel consumption rate at maximum capacity is $\alpha^*$, and the fuel consumption rate in the unloaded state is $\alpha_0$, then based on formula one, the values will be obtained in accordance with relations two and three.

$$\alpha_0 = \omega\theta_c + b \tag{2}$$

$$\alpha^* = \omega(\theta_c - Q_t) + b \tag{3}$$

Therefore, $\gamma$ will be obtained using Equation (4), as follows:

$$\omega = \frac{\alpha^* - \alpha_0}{Q_t} \tag{4}$$

Similarly, if $\alpha(\theta_u)$, we can write Equation (5) as follows:

$$\alpha_{(\theta_u)} = \alpha_0 + \frac{\alpha^* - \alpha_0}{Q_t}\theta_u \tag{5}$$

According to Equations (4) and (5), the rate of consumption (in terms of fuel units per kilometer) and the amount of fuel consumed (in terms of fuel units) are estimated in accordance with Equations (6) and (7) in order to move cargo with the weight of $F_{ij}$ from node $i$ to $j$.

$$\alpha_{cg}(F_{cg}) = Max\left[\alpha_0 + \frac{\alpha^* - \alpha_0}{Q_t}(F_{cg}), 0\right] \tag{6}$$

$$fp_{cg}^h = d_{cg}p_{cg}^t(F_{cg}) \tag{7}$$

Introduction of the Objective Functions

In this model, the objective function minimizes the sum of the significant failure costs in the distribution process, which includes two parts. The total failure cost comprises the cost of food stacked on top of each other during transportation, the cost of damage caused by the vehicle stopping during the delivery of customers' orders, the cost of creating warehouses, the cost of using cars, the cost of transportation, and the vehicle's refrigeration cost. Determining costs associated with the refrigeration of food to prevent spoilage depends on two factors, as follows:

✓ Heat transfer inside and outside the refrigerator and freezer due to temperature differences during transportation time.

✓ Heat exchange due to air convection during the loading and unloading time. The vehicles' cooling system costs can be obtained by calculating the energy consumption for refrigeration.

✓ Regarding the difference in temperature between the outside and inside of the refrigerator, the thermal load can be obtained using Equation (8).

$$\Gamma_\tau = (1 + \alpha)\delta\sqrt{\sum S_w \sum S_s}\Delta T \tag{8}$$

The thermal load of the vehicle during the loading/unloading time can be obtained using Equation (9).

$$\Gamma_\sigma = (0.54 L_V + 3.22)\Delta T \beta \tag{9}$$

where $\beta(0.25,2)$ is related to the frequency of opening the refrigerator door, thus enabling its value to be obtained.

Therefore, the Objective Function (OF) of the problem will be defined according to Equation (10).

$$
\begin{aligned}
MinZ = &\sum_{c \in C}\sum_{g \in v'}\sum_{t \in T}\sum_{p \in P}\theta_p TR_{cg}q_{pc}X_{cgt} + \sum_{c \in C}\sum_{g \in G}\sum_{t \in T}\sum_{p \in P}\mu_p S(g)q_{pc}pr_p X_{cgt}\sum_{g \in G}F_g M_g + \sum_{t \in T}\sum_{c \in C}\sum_{g \in G}CF_t X_{cgt} \\
&+ \sum_{c \in C'}\sum_{g \in V'}\sum_{t \in T}C^t\left(\alpha_0^t X_{cgt} + \frac{\alpha_t^* - \alpha_0^t}{Q_t}(F_{cg})\right)d_{cg} + \sum_{c \in C}\sum_{g \in V'}\sum_{t \in T}(1 + \omega_t)\delta\sqrt{\sum S_w \sum S_n}\Delta T CR^t TR_{cg}X_{cgt} \\
&+ \sum_{c \in C}\sum_{g \in G}\sum_{t \in T}(0.54 l_v + 3.22)\Delta T \beta s_g CR^t X_{cgt}
\end{aligned}
\tag{10}
$$

$$\sum_{t \in T}\sum_{g \in V'}X_{cgt} = 1 \forall c \in C \tag{11}$$

$$\sum_{t \in T}\sum_{g \in V'}X_{cgt} \leq 1 \forall c \in F' \tag{12}$$

$$\sum_{g \in V'}F_{cg} - \sum_{g \in V'}F_{cg} = \sum_{p \in P}q_{cp} \forall c \in V \tag{13}$$

$$\sum_{g \in V'}X_{cgt} = \sum_{g \in V'}X_{cgt} \forall c \in V', t \in T \tag{14}$$

$$F_{cg} \leq \sum_{t \in T}Q^t X_{cgt} \forall c \in C, g \in V' \tag{15}$$

$$\sum_{g \in V}F_{gt} = \sum_{g \in V}\sum_{p \in P}W_{gt}U_{pc}O_p \forall t \in T, c \in C \tag{16}$$

$$\sum_{g \in V}F_{gc} = 0 \forall c \in C \tag{17}$$

$$F_{cg} \leq \sum_{t \in T}\left(Q_t - \sum_{p \in P}q_{cp}O_p\right)X_{cgt} \forall c \in V, g \in V' \tag{18}$$

$$F_{cg} \geq \sum_{p \in P}U_{cp}O_p \sum_{t \in T}X_{cgt} \forall c \in V', g \in V \tag{19}$$

$$\sum_{c \in V}\sum_{p \in P}U_{cp}O_p W_{cg} \leq Q_g M_g \forall g \in G \tag{20}$$

$$\sum_{c \in C}\sum_{g \in G}X_{cgt} = 1 \forall t \in T \tag{21}$$

$$\sum_{t \in T} X_{tc} \leq W_{tc} \forall c \in C \tag{22}$$

$$\sum_{t \in T} X_{cgt} \leq W_{ct} + \sum_{g \in G, g \neq t} W_{gt} \leq 2 \forall t \in T, (c,g) \in C, c \neq g \tag{23}$$

$$\left( \rho_0^t X_{cgt} + \frac{\rho_t^* - \rho_0^t}{Q^t}(F_{cg}) \right) d_{cg} - (1 - X_{cgt})B^t \leq y'_{ct} - y''_{gt} \leq \left( \rho_0^t X_{cgt} + \frac{\rho_t^* - \rho_0^t}{Q^t}(F_{cg}) \right) d_{cg} + (1 - X_{cgt})B^t$$
$$\forall c \in V', t \in T, g \in F' \tag{24}$$

$$y'_{ct} \geq \left( \rho_0^t X_{cgt} + \frac{\rho_t^* - \rho_0^t}{Q^t}(F_{cg}) \right) d_{cg} \forall c \in V, t \in T, g \in G \tag{25}$$

$$y'_{ch} = B^h \forall h \in H, c \in V'' \tag{26}$$

$$B^t - \left( \alpha_0^t X_{cgt} + \frac{\alpha_t^* - \alpha_0^t}{Q^t}(F_{cg}) \right) d_{cg} \geq y'_{gt} \forall c \in V'', t \in T, g \in V' \tag{27}$$

$$EST_c \leq \tau_c^t + S_c \leq LST_c \forall c \in C, t \in T \tag{28}$$

$$\tau_c^t + (S_c + t_{cg}) X_{cgt} - \tau_c^t \leq \propto (1 - X_{cgt}) \forall c \in C, g \in V', t \in T \tag{29}$$

$$\tau_g^t \geq \tau_c^t + t_{cg} X_{cgt} + r_t (B^t - y''_{ct}) X_{cgt} - (\propto + r_t B^t)(1 - X_{cgt}) \forall c \in F', \forall g \in V', t \in T \tag{30}$$

$$\sum_{c \in V'} \sum_{g \in F'} X_{cgt} \leq 1 \forall t \in T \tag{31}$$

$$X_{cgt} \in \{0,1\} \forall c, g \in V', t \in T, c \neq g \tag{32}$$

$$W_{cg} \in \{0,1\} \forall c, g \in V', c \neq g \tag{33}$$

$$X'_{cgt} \in \{0,1\} \forall c \in F', \forall g \in V', t \in T, c \neq g \tag{34}$$

$$M_g \in \{0,1\} \forall g \in G \tag{35}$$

$$\tau_g^t \rangle 0 \forall t \in T, g \in V' \tag{36}$$

$$y'_{gt} \rangle 0 \forall t \in T, g \in V' \tag{37}$$

$$y''_{gt} \rangle 0 \forall t \in T, g \in V' \tag{38}$$

In this model, constraint (11) indicates that each customer belongs to precisely one route, which is used only once. Constraint (12) guarantees that each refueling station is visited only once. Constraint (13) indicates that the demand of each customer is satisfied. Constraint (14) indicates that the entrance to each vertex equals the number of exits from that vertex. Constraint (15) indicates that the total goods loaded at each edge should not exceed the vehicle's capacity moving on that edge. Constraint (16) indicates the capacity of each warehouse and ensures that the goods stored in each warehouse satisfies the demand of customers assigned to that warehouse. Constraint (17) indicates the number of goods remaining in the vehicle when it returns to the warehouse, which is equal to zero.

Constraints (18) and (19) represent the capacity of the vehicles. Constraint (20) guarantees that the total demand supplied by warehouses does not exceed the capacity of each warehouse. Constraint (21) ensures that each customer is assigned to only one warehouse and vehicle. Constraints (22) and (23) are related to the sub-tour elimination. Constraints (24) and (25) indicate the conditions for finding the base fuel level of vehicles in successive vertices. Constraint (26) represents the reduced ability to charge fuel tanks to maximum capacity at refueling stations and warehouses. Constraint (27) represents the battery level (fuel tank), which is equal to the fuel tank's maximum capacity; however, the energy requirement reduces the maximum capacity in the corresponding edge. Constraints (28)–(30) guarantee that the customers' time window is not violated. Constraint (31) is related to the fuel stations and vehicles, indicating that each vehicle can only refuel once. Constraints (32)–(38) show the decision variables in the model.

### 3.2. Linearization of the Mathematical Model of the Problem

The mathematical model of the mixed-integer programming (MIP) problem, presented in this research, is valid in the form of relations (10) to (38), but upon reflection of Constraint (30), this constraint was found to be nonlinear after multiplying the two decision variables $X_{cgt}$, $y''_{ct}$ together; this made the model nonlinear. Constraint (30) thus becomes Constraints (39)–(42) in order to linearize the model, and it is enough to replace these constraints with Constraint (33) from the previous model.

$$y''_{ct} - X'_{cgt} \leq M - MX_{cgt} \forall c \in F', \forall g \in V', t \in T \tag{39}$$

$$X'_{cgt} \leq y''_{ct} \forall c \in F', \forall g \in V', t \in T \tag{40}$$

$$X'_{cgt} \leq MX_{cgt} \forall c \in F', \forall g \in V', t \in T \tag{41}$$

$$\tau^t_g \geq \tau^t_c + t_{cg}X_{cgt} + r_t B^t X_{cgt} - r_t X'_{cgt} - (M + r_t B^t) \forall c \in F', \forall g \in V', t \in T \tag{42}$$

## 4. Solution Approaches

In the literature, the classic LRP is considered to be one of the hard problems, and finding the optimal solution in this field is difficult, even in relatively small dimensions. Moreover, this problem is complex for large problems, and almost impossible using exact methods, as per Garey and Johnson (1979) [55]. For this reason, innovative and meta-heuristic methods have been developed in most studies to solve these problems. In this section, meta-heuristic algorithms used in this research, the genetic algorithm, and the simulated annealing algorithm, will be briefly described; then, the efficiency of these algorithms will be evaluated to solve the introduced mathematical model.

### 4.1. Genetic Algorithm

A genetic algorithm is a search technique in computer science that aims to find the optimal solution for complex optimization problems. This algorithm is one of the evolutionary algorithms inspired by biological science, incorporating factors such as inheritance, mutation, sudden selection, natural selection, and combination. The genetic algorithm is more efficient at solving discrete and nonlinear problems [56]. First, the answer is shown to be a 1*g matrix with zero and one rows. One indicates that the warehouse will be built, and zero indicates that the warehouse will not be built. For example, consider g = 3, the matrix for which is randomly generated in Table 5, and warehouses two and three are built in the chromosome mentioned above.

**Table 5.** Chromosome with a number of genes that determine the number of warehouses.

| Warehouse | $j_1$ | $j_2$ | $j_3$ |
|---|---|---|---|
| $X_1$ | 0 | 1 | 1 |

The second part of the answer shows the allocation of customers to each of the warehouses. This chromosome is a 1*N matrix whose entries are filled with integers, indicating that each warehouse is assigned to a customer. For example, consider that the number of customers equals six, and warehouses two and three have been built; allocating customers to warehouses would thus occur as shown in Table 6.

**Table 6.** Chromosome with a number of genes that determine how customers are allocated to warehouses.

| Customer | $C_1$ | $C_2$ | $C_3$ | $C_4$ | $C_5$ | $C_6$ |
|---|---|---|---|---|---|---|
| $X_2$ | 2 | 1 | 3 | 3 | 2 | 3 |

The third part of the answer relates to the sequence of customer visits. This matrix has dimensions 1*N, and its fields are filled with random sequences of numbers between 1 and N. This matrix is shown in Table 7.

**Table 7.** Chromosome with a number of genes that determine the sequence of meeting customers with cars.

| Customer | $C_1$ | $C_2$ | $C_3$ | $C_4$ | $C_5$ | $C_6$ |
|---|---|---|---|---|---|---|
| $X_3$ | 1 | 2 | 3 | 6 | 4 | 5 |

Based on this part of the answer, it is determined how the meeting sequence takes place, with regard to customers who have been assigned to a warehouse. For example, based on matrix $X_2$ (Table 6), it was determined that customers 1, 4, and 5 were assigned to warehouse two. Therefore, based on the $X_3$ matrix, it may be determined that the order in which these customers should be met by vehicle is 4-5-1.

The fourth part of the answer concerns the sequence where vehicles are allocated to warehouses. This matrix has dimensions T*1, and its chromosome is shown in Table 8. Supposing H is equal to 2, the matrix is generated as follows. Customers are first assigned to vehicle two when routing. Then, after considering restrictions (for example, each vehicle is allocated to a warehouse or vehicles have a capacity limit), customers are allocated to the first vehicle.

**Table 8.** Chromosome with a number of genes that determine the allocation of cars.

| Customer | $C_1$ | $C_2$ |
|---|---|---|
| $X_4$ | 2 | 1 |

The fifth part of the answer shows the sequence of fuel stations, which is a 1*F matrix, it shows which fuel stations are allocated to vehicles, and in which order; this chromosome is shown in Table 9. Supposing F equals 1, this matrix is randomly generated as follows.

**Table 9.** Chromosome with a number of genes that determine the allocation of fuel stations to cars.

| Customer | $C_2$ | $C_3$ | $C_4$ | $C_5$ |
|---|---|---|---|---|
| $X_5$ | 3 | 2 | 5 | 2 |

In the presented model, the initial population is generated in the form of a sieve. To produce each member of the population, warehouses are created randomly, and the distances between each customer and the warehouses are measured. Then, while considering the limited capacity of the warehouses, customers are assigned to the closest warehouses. After allocating customers to warehouses, one vehicle is randomly selected from each warehouse, along with the customer who is closest to the warehouse (provided that the delivery can occur within the time window and the nearest fuel station is on route). There should be sufficient fuel to navigate the distance between the warehouse and the customer and vice versa. The first customer is selected based on closest distance to the warehouse; then, the customer with the closest distance to the first customer is selected. This process continues until all customers assigned to the warehouses are served. To ensure that the answers are justified in the algorithm, first, all the random answers generated in the semi-code, as defined in the problem's limits, are tested; then, the justified answers are accepted and the unjustified answers are removed from the set of answers.

### 4.1.1. Crossover

In this algorithm, one-point crossover is used. The operation of crossing to produce a new generation is randomly applied to two selected parents that are related to one of the five parts of the answer. For example, the operation of one-point crossover is shown in Figure 3 for two parent chromosomes that are related to allocating customers to warehouses; in other words, the second part of the answer display is shown to produce children.

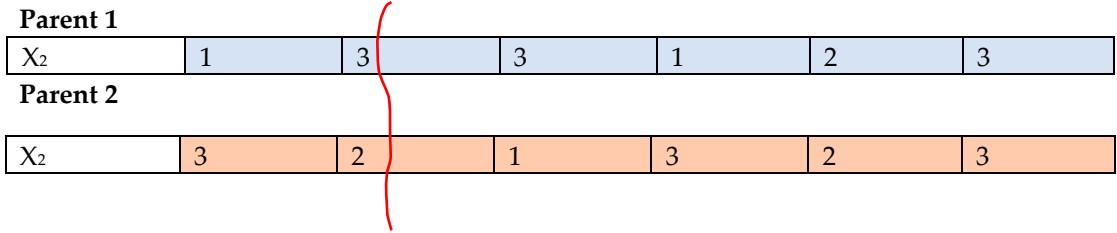

**Figure 3.** One-point crossover action on the chromosome in the third gene.

At each point in the crossover process, a gene from two chromosomes, corresponding to one of the parts of the answer display, is randomly selected, and the crossover action is performed on that gene. The example in Figure 4 assumes point 2 to be the crossover point in the second part of the answer display; the produced chromosomes, also shown in Figure 4, shows the allocation of customers to warehouses.

**Child 1**

| $X_2$ | 2 | 3 | 1 | 3 | 2 | 1 |
|---|---|---|---|---|---|---|

**Child 2**

| $X_2$ | 3 | 1 | 2 | 2 | 1 | 3 |
|---|---|---|---|---|---|---|

**Figure 4.** Chromosomes are produced using one-point crossover.

### 4.1.2. Mutation

The mutation operator increases the dispersion of the answers, and the search space is further investigated. The presented algorithm uses all three types of mutation operators, as follows: swap, inversion, and insertion. Therefore, in each iteration of the algorithm, one of these operators is randomly applied to each part of the answer display.

### 4.1.3. The Stopping Condition

The stopping condition in the presented algorithms reach a certain number of repetitions.

### 4.2. Simulated Annealing (SA) Algorithm

The annealing simulation algorithm is a simple and effective meta-heuristic optimization algorithm for solving optimization problems. In 1983 and 1985, Kirkpatrick et al. and Cerny, respectively, used the annealing simulation algorithm for other optimization problems. The main advantage of the annealing simulation algorithm is its ability to solve problems at the local optimal point while moving toward the optimal point [57,58].

Neighborhood Structure

To create a neighborhood in this algorithm, three efficiency operators are used, as follows: 1-swap, 2-inversion, and 3-insertion. They are used for each of the five parts of the answer display, as mentioned in previous sections. For this purpose, a random number like P, between 0 and 1, is generated, and then, based on that, changes are applied to one of the parts of the answer display using the mentioned operators, as shown in Table 10. The way operators are chosen to create a change and a new neighborhood in each iteration on each component of the answer display is completely random.

**Table 10.** Applying neighborhood-building operators to the components of the answer display.

| Changes Applied to the Answer Display Sub-Section | Randomly Generated Number (P) |
| --- | --- |
| Allocation of warehouses and customers | $0 < P \leq 0.25$ |
| The sequence of meeting customers | $0.25 < P \leq 0.50$ |
| Allocation of cars | $0.50 < P \leq 0.75$ |
| Meeting at fuel stations | $0.75 < P \leq 1$ |

With the creation of new neighborhoods, the conditions for improving within and between the grid are provided, and the algorithm is able to find more search opportunities in the solution space. For example, if the random number generated at the beginning of the neighborhood creation process for a specific iteration of the algorithm is equal to P = 0.20, and the matrix represents a sequence for meeting customers with an initial answer or an accepted answer of the previous iteration of the algorithm, then the neighborhood structure created by applying each of the operations will be one of those shown in Figures 5–8.

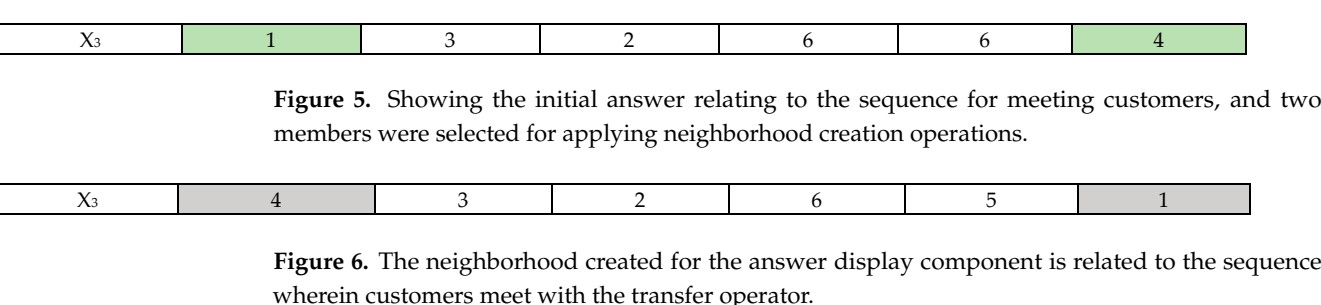

**Figure 5.** Showing the initial answer relating to the sequence for meeting customers, and two members were selected for applying neighborhood creation operations.

| $X_3$ | 4 | 3 | 2 | 6 | 5 | 1 |

**Figure 6.** The neighborhood created for the answer display component is related to the sequence wherein customers meet with the transfer operator.

| $X_3$ | 1 | 5 | 6 | 2 | 3 | 4 |

**Figure 7.** The neighborhood created for the answer display component is related to the sequence wherein customers meet with the inversion operator.

| $X_3$ | 1 | 4 | 3 | 2 | 6 | 5 |

**Figure 8.** The neighborhood created for the answer display component is related to the sequence wherein customers meet with the insertion operator.

## 5. Validation of the Mathematics Model for Metaheuristic Algorithms

In this section, the proposed mathematics model is first validated by solving a small-sized numeric example using LINGO 17.0 software and analyzing the gained results.

Then, more numeric examples are solved, and the efficiency of the genetic algorithm and simulated annealing is reviewed. In this article, 46 numerical examples are produced and placed into three categories, as follows: small, medium, and large. We tried to use data taken from the literature. The products delivered to customers fall into two types, and the characteristics of each product are presented in Table 11.

**Table 11.** Information related to each type of frozen food.

| References | The Type of Product | | Parameter |
|---|---|---|---|
| | **Y1** | **Y2** | |
| Zheng and Chen (2014) [35] | USD 20 | USD 75 | $P_\gamma$ |
| Zheng and Chen (2014) [35] | 0.070% | 0.2% | $\mu_\gamma$ |
| Zheng and Chen (2014) [35] | 0.04% | 0.8% | $\theta_\gamma$ |
| Wang et al. (2018) [53] | 10 Kg | 10 Kg | $O_\gamma$ |

To determine geographic location and customer demand, as well as the coordinates of potential distribution centers, we used data taken from the article by Wang et al. (2016), relating to the MPF Logistics Company in China; this company is active in the field of warehousing and the distribution of cold foods such as dairy products and frozen meat. This center has 60 customers and 0 potential distribution centers in the main area, with a total capacity of 700 tons. At first, a number is generated randomly with a uniform distribution in the interval (4, 40); this is the earliest service time for customers. Then, a number is randomly generated in the interval (1, 10); this is the latest service time for customers. The duration of customer service is also generated as a random number with a uniform distribution in the interval (0.04, 0.60). The speed of movement for all types of cars is fixed and equal to 400 km/h, six vehicles were used, and their other specifications are shown below. The cost of building each of the depots varies for each of the examples, and the interval is random, with a uniform distribution between (200, 4200). Other data are assumed to be constant for all examples, as shown in Table 12.

**Table 12.** Information related to each vehicle.

| Parameter | Value |
|---|---|
| $r_h$ | U(0.0083; 0.04) |
| $\rho_0^h$ | U(0; 1) |
| $\rho_h^*$ | U(0.1; 2) |
| $C^h$ | U(1, 100) \$/lit |
| $CF_h$ | U(10; 500) |
| $CR^h$ | U(0.05; 1) \$ |
| $Q^h$ | U(100; 1000) kg |
| $B^h$ | U(5; 100) |

### 5.1. Validation of the Mathematical Model and Solving A Small Numerical Example

For this purpose, first, a small numerical example including two customers, two potential distribution centers, four fuel stations, and two cars was produced. It noted the considerations defined in Sections 1 and 2, and then, in order to validate and demonstrate the capability of the model, the problem was divided into two scenarios. The capacity of the fuel tanks of cars, according to the values mentioned in Tables 13 and 14, was considered.

**Table 13.** Data related to the refrigeration system and other parameters of the problem.

| Parameter | Value |
|---|---|
| $\beta$ | (0; 1) |
| $\varepsilon$ | (0; 1) |
| $\Delta T$ | (1, 50) |
| External dimensions of the refrigerator | $496 \times 172 \times 246$ cm |
| Internal dimensions of the refrigerator | $280 \times 155 \times 154$ cm |
| $\varepsilon$ | Kcal2.49 |
| $L_v$ | 0.08 |

**Table 14.** The results of the exact solution of the small numerical example, obtained using Lingo.

| Algorithm Gap Rate | | | The Results of Each Solution Method | | | | | | Number of Fuel Stations | Number of Vehicles | | Number of Customers | Number of Potential Depot Centers | Problem Number | Problem Class |
|---|---|---|---|---|---|---|---|---|---|---|---|---|---|---|---|
| | | | Genetic | | Simulated Annealing | | Lingo | | | Kind 1 | Kind 2 | | | | |
| Genetic to Simulated Annealing. | Genetic to Lingo | Simulated Annealing to Lingo | Solving Time (Seconds) | The Optimal Answer | Solving Time (Seconds) | The Optimal Answer | Solving Time (Seconds) | The Optimal Answer | | | | | | | |
| %−2.0 | %−2.1 | %+0.5 | 54 | 172.4 | 6 | 1914.2 | 2 | 1904.2 | 4 | 1 | 1 | 6 | 3 | 2 | |
| %−1.5 | %+1.7 | %+3 | 87 | 190.8 | 7 | 1034.5 | 15 | 1074.8 | 2 | 1 | 1 | 8 | 3 | 2 | |
| %−1 | %−3.6 | %−2.8 | 108 | 289.0 | 9 | 3114.1 | 144 | 2174.6 | 4 | 2 | 1 | 10 | 4 | 2 | Small |
| %−2.0 | %−3.1 | %+3.4 | 191 | 407.4 | 19 | 2323.3 | 1955 | 3245.7 | 5 | 2 | 1 | 9 | 4 | 5 | |
| %−2.8 | %−1.4 | %+1.2 | 206 | 301.5 | 17 | 2271.9 | 1422 | 2241.8 | 4 | 2 | 1 | 12 | 3 | 1 | |
| %−0.5 | %−1.5 | %−1.1 | 260 | 167.0 | 22 | 5321.9 | 17,150 | 2347.9 | 4 | 2 | 1 | 15 | 4 | 6 | |
| %−1.5 | %−2.4 | %−0.9 | 259 | 2457.2 | 25 | 4495.5 | 3921 | 2019.0 | 4 | 2 | 2 | 15 | 4 | 8 | |
| %−5.1 | - | - | 357 | 2708.3 | 31 | 2853.5 | - | - | 3 | 3 | 1 | 50 | 4 | 8 | Middle |
| %−10.2 | - | - | 372 | 279.1 | 37 | 3230.1 | - | - | 4 | 2 | 2 | 20 | 4 | 9 | |
| %−15 | - | - | 807 | 8926.7 | 80 | 1057.7 | - | - | 10 | 1 | 4 | 55 | 4 | 13 | |
| %−10.2 | - | - | 1855 | 101.8 | 170 | 11,290.5 | - | - | 8 | 6 | 4 | 45 | 5 | 11 | |
| - | - | - | 2874 | 173.6 | 222 | 1330.6 | - | - | 9 | 6 | 5 | 45 | 5 | 10 | Large |
| - | - | - | 4862 | 153.5 | 334 | 1740.2 | - | - | 6 | 10 | 3 | 55 | 7 | 12 | |
| - | - | - | 5150 | 142.0 | 339 | 1653.6 | - | - | 10 | 5 | 4 | 60 | 5 | 14 | |
| - | - | - | 5156 | 153.1 | 233 | 15,655.3 | - | - | 7 | 11 | 6 | 60 | 5 | 15 | |

Reducing the capacity of the fuel tanks of electric and combustion vehicles to 20 and 10 fuel units, respectively, was coded and solved using Lingo software, and the results obtained from solving each scenario are shown in below. In the following section, the level of fuel consumption is assumed to occur at a constant rate, and at a rate that is equal to the average fuel consumption rate; fuel consumption is also reviewed when the vehicle is unloaded and at full capacity.

In scenario A, despite the restrictions related to consumption and the level of fuel in the tanks of cars, due to the adequate level of fuel in the tanks of cars that are in the process of serving customers, there is no need refuel any of the cars at fuel stations. By reducing the capacity of the fuel tanks of the cars in scenario B, car number two (veh2) travels to refueling station number three (f3) in order to continue the route, and after charging and bringing the fuel level of the tank to its maximum capacity, it continues its journey. If there was no possibility of refueling cars in scenario B, the problem would have no optimal solution. The costs of the distribution process of each scenario are shown in Figures 9 and 10.

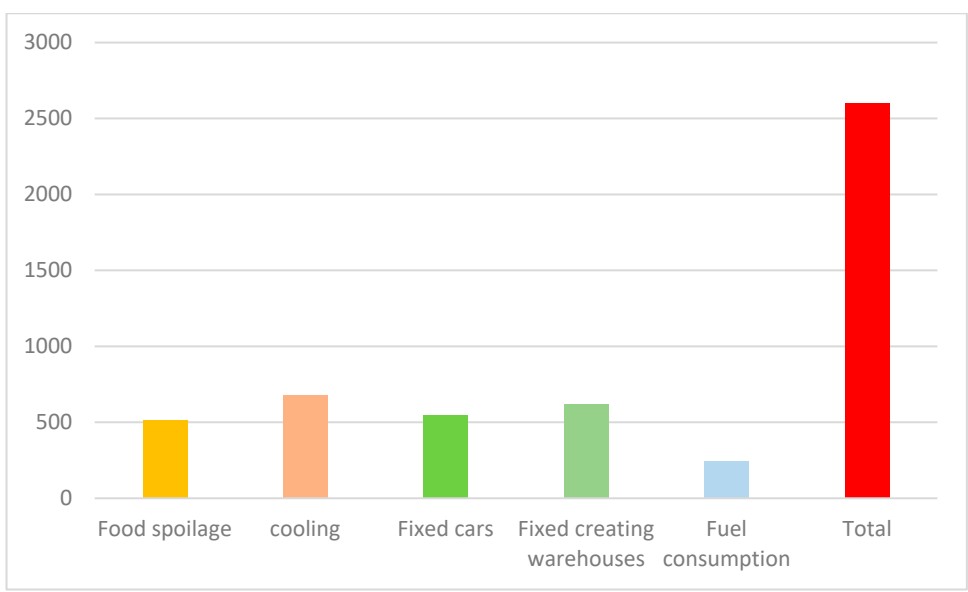

**Figure 9.** Costs of the distribution process of scenario A.

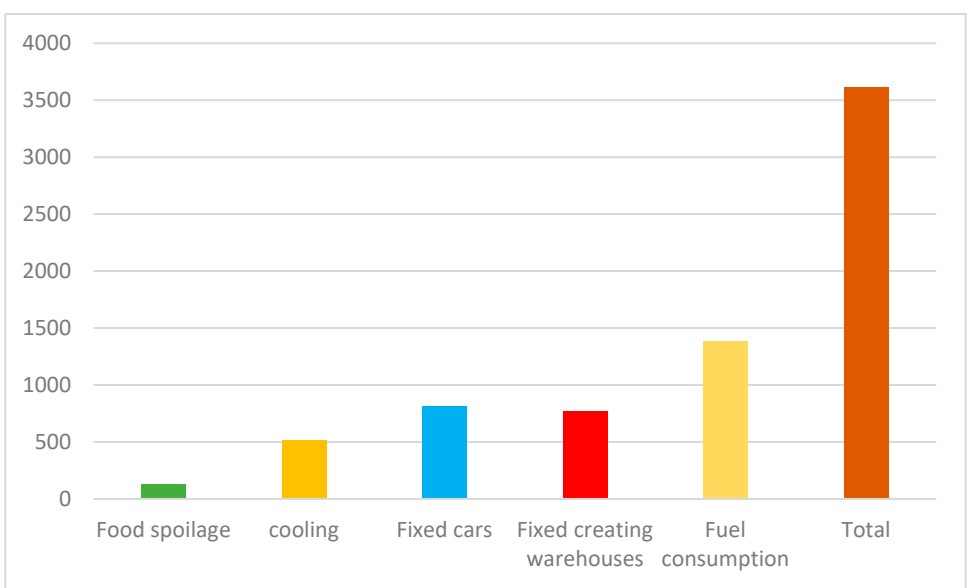

**Figure 10.** Costs of the distribution process of scenario B.

In the following section, the fuel consumption of each of the scenarios was evaluated and analyzed based on the assumption that fuel consumption occurs at a constant rate and that it is equal to the average fuel consumption rate; fuel consumption was measured when the vehicle was unloaded and at capacity, as shown Table 15. The results confirm an average saving of 42.2% in fuel consumption, compared with the existing models which have a fixed fuel consumption rate.

**Table 15.** Comparison of the fuel consumption of the proposed model compared with other models that have a fixed fuel consumption rate.

| Scenario | Vehicle Number | Fuel Consumption Unit | Fixed Fuel Consumption Rate | Total Engine Mileage (km) | Amount of Fuel Consumed | | The Amount by which Fuel Consumption Increased/Decreased in the Proposed Model Compared with other Models (%) |
|---|---|---|---|---|---|---|---|
| | | | | | Proposed Model | Common Models With Fixed Fuel Consumption Rates | |
| A | Veh1 | Jules /km | 1.6 | 218.6 | 111.7 | 252.8 | −9% |
| | Veh2 | lit/km | 0.241 | 50 | 11.30 | 10.72 | −26.11% |
| B | Veh1 | Jules /km | 1.5 | 14.2 | 12.04 | 21.4 | −20.3% |
| | Veh2 | lit/km | 0.270 | 209.4 | 59.03 | 70.43 | −19.9% |
| | | Total | | | 367.44 | 47.40 | −12.8% |

### *5.2. Setting the Parameters of Meta-Heuristic Algorithms*

To adjust the parameters of the proposed solution algorithms in this research, the Taguchi test method was used, and the results of this test were reported as average effects using Minitab version 17.1 software for each of the algorithms. Finally, the optimal levels of each of the parameters of the algorithms were determined.

To validate the proposed algorithms, the solution for these algorithms, regarding small-size problems, will be compared with the optimal solution obtained from the Lingo 17.0 software. Regarding the reported numerical examples, problems which required processing times of more than 3 h, using the Lingo software, are omitted. The number of repetitions of algorithm loops after multiple executions is experimentally relative to the resulting convergence; to obtain a relatively suitable answer in a reasonable solution time, the required number of repetitions is considered to be 200. Each problem was solved five times with each of the algorithms on a personal computer with 21.2 GHz Intel, 2 GB RAM Core5, and the best solution is shown in the computational results, displayed in Tables 16 and 17. Regarding the solved numerical examples with small dimensions, the maximum deviation of the refrigeration and genetic simulation algorithm, compared with the solution obtained from the exact solution, is 2.1% and 2.6%, respectively. Due to the insignificant number of deviations, the performance of both algorithms for solving the proposed model is acceptable. However, by comparing the answers obtained from each algorithm, we found that the genetic algorithm provided better answers than the refrigeration simulation algorithm. To increase the validity of the proposed algorithms, in addition to comparing the answers and the obtained solution times, the paired t-test was used. First, we tested the hypothesis of improvement using the performance of the genetic algorithm, which was compared with the improvement of the refrigeration simulation algorithm in terms of solution and solution time. For this purpose, two hypothesis 1 and hypothesis zero are as follows: the answer to the refrigeration simulation algorithm is equal to the genetic algorithm, or conversely, the solution to the refrigeration simulation algorithm is more significant than the genetic algorithm; and the solution time of the genetic algorithm is equal to that of the refrigeration simulation algorithm.

**Table 16.** The set parameters of meta-heuristic algorithms.

| Algorithm Type | The Name of the Parameter | Limits of Tested Values of Parameters | | | Confirmed Values of Parameters from the n Taguchi Test |
|---|---|---|---|---|---|
| | | Lower | Middle | Upper | |
| Genetic | Number of primitive Population | 60 | 80 | 100 | 100 |
| | Intersection rate | 65% | 75% | 85% | 65% |
| | Mutation rate | 20% | 30% | 40% | 40% |
| Simulated Annealing | Initial temperature | 80 | 100 | 120 | 100 |
| | Rate of reduction in terms of temperature | 65% | 80% | 99% | 65% |

**Table 17.** Data sources for the parametric settings of the meta-heuristic algorithm.

| Sources |
|---|
| EDR Santibanez Gonzalez et al. (2023) [34] |
| Goodarzian et al. (2023) [59] |
| Hasan et al. (2023) [60] |

The parameter settings of the meta-heuristic algorithm were taken from different literature sources, which we have listed in Table 18.

**Table 18.** Comparative results from solving numerical examples of the model with meta-heuristic algorithms.

| Scenario | Vehicle Number | tank Capacity/ | Load Carrying Capacity(k) | Tour Sequence Node c | Tour Sequence Node g | The Distance between Two Nodes (km) | The Amount of Cargo Transported between Two Nodes | Fixed | Variable | Total | The Amount of Fuel Consumed between Two Nodes (Fuel Unit) | The Remaining Fuel Level of the Tank/Battery at the Moment of Entering Node j | The Moment of Entering Node j | The Value of the Objective Function |
|---|---|---|---|---|---|---|---|---|---|---|---|---|---|---|
| A | Veh1 | 34,000 | 3560 | C1 | G2 | 11.8 | 3760 | 1.2 | 0.901 | 1.814 | 51.3 | 243.7 | 2.56 | 198.90 |
| | | | | C3 | G1 | 34.5 | 2950 | | 0.651 | 1.632 | 35.1 | 248.6 | 3.35 | |
| | | | | C4 | G8 | 100.6 | 1520 | | 0.473 | 1.143 | 51.2 | 247.3 | 7.91 | |
| | | | | C8 | G5 | 99.4 | 1091 | | 0.202 | 1.304 | 46.2 | 24,811 | 8.6 | |
| | | | | C3 | G6 | 7.7 | 670 | | 0.158 | 43.6 | 247.8 | 11.75 | | |
| | | | | C10 | G2 | 49.2 | 12.6 | | 0.1 | 1.2 | 49.2 | 248.2 | 12.49 | |
| | Veh2 | 52 | 9073 | C1 | G7 | 80.7 | 2880 | | 0.249 | 0.213 | 3.2 | 56.8 | 3.61 | |
| | | | | C5 | G10 | 21.6 | 1610 | 0.105 | 0.031 | 0.205 | 4.3 | 52.5 | 4.43 | |
| | | | | C1 | G5 | 14.9 | 350 | | 0.088 | 0.189 | 2.5 | 49.8 | 9.22 | |
| | | | | C9 | G2 | 6.8 | 15.4 | | 0.01 | 0.11 | 1.1 | 48.7 | 11.07 | |
| B | Veh1 | 43 | 3560 | C8 | G5 | 25.8 | 750 | 1.5 | 0.219 | 1.211 | 8.24 | 20.76 | 9.37 | 26.47 |
| | | | | C2 | G5 | 6.8 | 10.1 | | 0.18 | 1 | 6.80 | 14.90 | 10.55 | |
| | | | | C6 | C2 | 40.7 | 4200 | | 0.011 | 0.264 | 5.71 | 34.29 | 0.70 | |
| | | | | C2 | G4 | 44.9 | 2867 | | 0.061 | 0.092 | 7.25 | 18.72 | 2.56 | |
| | | | | C7 | G1 | 53.5 | 2251 | | 0.052 | 0.218 | 4.60 | 15.04 | 3.35 | |
| | Veh2 | 37 | 9073 | C9 | G5 | 60.6 | 1769 | 0.199 | 0.033 | 0.205 | 7.1 | 7.95 | 5.66 | |
| | | | | C8 | G3 | 77.7 | 1022 | | 0.025 | 0.180 | 6.04 | 1.91 | 6.32 | |
| | | | | C3 | G2 | 9.4 | 1050 | | 0.025 | 0.190 | 12.84 | 27.16 | 7.98 | |
| | | | | C7 | G6 | 8.7 | 500 | | 0.010 | 0.175 | 6.41 | 20.75 | 8.6 | |
| | | | | C6 | G2 | 10.3 | 10.1 | | 0 | 0.165 | 2.37 | 18.39 | 9.14 | |

Based on the results, the null hypothesis is rejected in both tests, and the opposite hypothesis is accepted. In other words, the response values of the refrigeration simulation algorithm are larger than the genetic algorithm, and this issue can also be seen in the palette box diagram. Conversely, the solving time of the genetic algorithm is larger than that of the refrigeration simulation algorithm.

*5.3. Managerial Insights*

The paper proposed a location–routing model for a cold supply chain in emergency conditions to minimize relief cost, traveling time, and $CO_2$ emissions. The paper also considers the impact of COVID-19 on the supply chain network and relief logistics. The managerial insights of the paper are reported in before, which includes the proposed model's effectiveness in terms of minimizing relief costs, traveling time, and $CO_2$ emissions. The paper also highlights the importance of considering the impact of COVID-19 on the supply chain network and relief logistics. The proposed model can help decision makers design a sustainable cold supply chain for use in emergencies while considering the impact of the COVID-19 disaster.

## 6. Conclusions and Future Suggestions

This paper proposes a new mathematical model for location–routing problems in a sustainable cold supply chain while considering the COVID-19 disaster. The proposed model aimed to minimize the total cost of the supply chain, including transportation, inventory, and facility costs, while ensuring that the cold chain requirements were met. We also considered the impact of the COVID-19 pandemic on the supply chain, and we proposed a contingency plan to mitigate the associated risks.

To solve the proposed model, we used two algorithms, the simulated annealing algorithm and the genetic algorithm. The results of the computational experiments showed that the proposed algorithm could find high-quality solutions in a reasonable amount of time.

In this study, to align the problem more closely with real-world conditions and to make it more practical, we aimed to minimize environmental and economic risks in the logistics process; a heterogeneous fleet that used unconventional fuels was employed, and the amount of fuel that was used depended on the size of the load between the vertices. This problem was first modeled by considering limitations and assumptions, and then, the exponential time complexity of solving the problem in medium and large dimensions was considered. Genetic algorithms and refrigeration simulations were developed to solve the proposed problem. To validate the presented mathematical model, first, a small numerical example was solved accurately using Lingo software; the model confirmed an average saving of 41% in terms of fuel consumption. In the next section, 15 numerical examples were generated and placed into three categories, small, medium, and large. The performance of the algorithms was then evaluated and tested to solve the presented model. The results show the genetic algorithm's high quality in terms of finding a suitable solution for the model, as compared with the refrigeration simulation algorithm, especially when the problem-solving time is less favorable.

Future research can build on the work presented in this paper in several ways. First, the proposed model can be extended to consider other types of disasters, such as natural disasters or terrorist attacks. Second, the model can be modified to include more realistic assumptions, such as time-varying demands and capacity constraints. Third, the proposed algorithm can be further improved by incorporating other meta-heuristic algorithms or developing more efficient solution techniques. Finally, the proposed model can be applied to real-world case studies to evaluate its practicality and effectiveness.

Due to the need to maintain the freshness of food in the cold distribution chain, and with the expansion of online shopping, along with the advancement of information and communication technology, using the satisfaction function to reduce the duration and number of tours, as well as the number of times the refrigerator door is opened and closed, is an avenue for future research. Problem-solving in the form of two goals with opposite goals is also suggested as a future field of development based on the current study.

Moreover, noting the necessity of fuel consumption to refrigerate food in the real-world of refrigerated vehicles and considering fuel consumption in this regard, coupled with understanding the amount of fuel required to travel the distance between nodes, can be a proposed area of future research.

**Author Contributions:** Conceptualization, S.A., M.T. and M.M.; Methodology, S.A.; Software, S.A. and M.T.; Validation, S.A. and I.V.; Formal analysis, S.A.; Investigation, S.A. and M.T.; Resources, M.M.; Supervising: I.V. All authors have read and agreed to the published version of the manuscript.

**Funding:** This research received no external funding.

**Institutional Review Board Statement:** Not applicable.

**Informed Consent Statement:** Not applicable.

**Data Availability Statement:** The data presented in this study are available on request from the corresponding author.

**Conflicts of Interest:** The authors declare no conflict of interest.

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
