# Peer review of "Designing the Location–Routing Problem for a Cold Supply Chain Considering the COVID-19 Disaster"

_sustainability, doi:10.3390/su152115490_

Round 1
Reviewer 1 Report
The article discusses a Position-Orientation Problem (LRP) model for the distribution network of multiple perishable food products in the Cold Supply Chain (CSC), where vehicles can refuel at filling stations during the COVID-19 disaster. The heuristic method is used to solve large-scale problems.
The article is at a level to contribute to the literature. The article can be published.
Author Response
see attached response

Reviewer 2 Report
This paper studies a location–routing problem for sustainable cold supply chain, where COVID-19 is considered. I have the following questions.
1. Although the authors mention the COVID-19, the paper does not provide a clear expression of its essential impact on the location-routing problem. Furthermore, from a global perspective, the impact of COVID-19 on logistics activities seems to be less important. What are the value points of this study?
2. The key motivation and innovation points of this study are unclear. Although the authors make a literature comparison in the literature review, the necessity of the factors considered was not clearly explained from the perspective of the research background, which resulted in the model setting only highlighting research differences but not indicating theoretical value.
3. The authors make the algorithm comparisons, but I think that's not enough. The author should provide more management insights for practical applications.
4. The writing and standardization of this paper need to be improved. For example, in line 283, p(m_u) should be error; in line 290, the description “according to relations four and five” is non-standard; in Eq.(10), what is the meaning of the dots between the multiplication symbols.
This paper studies a location–routing problem for sustainable cold supply chain, where COVID-19 is considered. I have the following questions.
1. Although the authors mention the COVID-19, the paper does not provide a clear expression of its essential impact on the location-routing problem. Furthermore, from a global perspective, the impact of COVID-19 on logistics activities seems to be less important. What are the value points of this study?
2. The key motivation and innovation points of this study are unclear. Although the authors make a literature comparison in the literature review, the necessity of the factors considered was not clearly explained from the perspective of the research background, which resulted in the model setting only highlighting research differences but not indicating theoretical value.
3. The authors make the algorithm comparisons, but I think that's not enough. The author should provide more management insights for practical applications.
4. The writing and standardization of this paper need to be improved. For example, in line 283, p(m_u) should be error; in line 290, the description “according to relations four and five” is non-standard; in Eq.(10), what is the meaning of the dots between the multiplication symbols.
Reviewer 3 Report
Title:
Designing the Location–Routing Problem for Cold Supply Chain Considering the COVID-19 Disaster
The author(s) are suggested to incorporate the following suggestions:
Question:
Table 7 represents the results. I would like to suggest that keep the number of vehicles and problem class data in start of the table and algorithm results such as gap rate should be at the end. I cannot understand about the gap rate of large instances, only “%” signs are given.
Question:
· The small class problem should be solved by exact method also and compared with the proposed one results.
· LINGO is a nice option; however, I would suggest authors to check on Gurobi optimizer.
· Mention the source of the small to large size problems data.
· The distance data is Euclidean or real time? Please explicitly explain.
Question:
What’s the purpose of “total” on the graph in figures 14 and 15?
Question:
Parametric settings for the met-heuristic are taken randomly or from any literature source? If, then please cite the source. Otherwise, the parametric tuning should be carried out with proper design.
Question:
Do cite the following papers to strengthen the literature.
· Li, Q., Lin, H., Tan, X., & Du, S. (2020). H∞ Consensus for Multiagent-Based Supply Chain Systems Under Switching Topology and Uncertain Demands. IEEE Transactions on Systems, Man, and Cybernetics: Systems, 50(12), 4905-4918. doi: 10.1109/TSMC.2018.2884510
· Li, J., Yang, X., Shi, V., & Cai, G. G. (2023). Partial centralization in a durable-good supply chain. Production and Operations Management. doi: https://doi.org/10.1111/poms.14006
· Xiao, Y., Zhang, Y., Kaku, I., Kang, R., & Pan, X. (2021). Electric vehicle routing problem: A systematic review and a new comprehensive model with nonlinear energy recharging and consumption. Renewable and Sustainable Energy Reviews, 151, 111567. doi: https://doi.org/10.1016/j.rser.2021.111567
· Huang, W., Wang, X., Zhang, J., Xia, J., & Zhang, X. (2023). Improvement of blueberry freshness prediction based on machine learning and multi-source sensing in the cold chain logistics. Food Control, 145, 109496. doi: https://doi.org/10.1016/j.foodcont.2022.109496
Reviewer 4 Report
Major revision is required. And my specific comments are shown below:
1- The authors must redesign the paper according to the Sustainability Journal Templet
2- The abstract is not attractive and needs improvement
3- please support the sentence" Results show that in this case....," in abstract by obtained results
4- It could be interesting to summarize the commented literature works in a table to have a clear comparison between all. This could also help precisely formulating the contribution of the paper with respect to previous works
5- The results section description must improve. I suggest converting the results table as "Table 10. Comparative results of solving numerical examples of the model with meta-heuristic algorithms ". to figures, etc. Also I recommend the authors to comparing the obtained results " Figure 14. Composition of costs of the distribution process of scenario A" and "Figure 15. Composition of costs of the distribution process of scenario B " with related works
6- Please add the contribution section
7- Results need a deeper discussion.
8- The " 7.Conclusion and future suggestions" section must be supported by the results
9- The English language of the paper need proofreading
Extensive editing of English language required
Round 2
Reviewer 2 Report
The author has made modifications and I have no further questions.
The author has made modifications and I have no further questions.
Author Response
see previous response
Reviewer 3 Report
After going to the revised manuscript, I feel that authors have adequately answered the concerns and the paper can be accepted.
Author Response
see previous response
Reviewer 4 Report
Extensive editing of English language required
Extensive editing of English language required
Author Response
The authors would like to thank the anonymous reviewers for the constructive comments and suggestions.
The English language of the paper needs proofreading.
Reply: We apologize for the untidy language and unintentional errors and mistakes in parts of the paper. Several errors and typos mistakes have been corrected. The entire submittal has now been reviewed by one native English speaker and has also been proofread by an editor. We hope any matters relating to the incorrect use of words have been resolved. If there are particular items that the reviewer has, we will be glad to address them.
Thank you for your consideration. Best Regards.